# Miniaturized and untethered McKibben muscles based on photothermal-induced gas-liquid transformation

Wenfei Ai [1,2,4], Kai Hou[1,4], Jiaxin Wu[1,2], Yue Long [1,3] ✉ & Kai Song [1,2,3] ✉

Pneumatic artificial muscles can move continuously under the power support of air pumps, and their flexibility also provides the possibility for applications in complex environments. However, in order to achieve operation in confined spaces, the miniaturization of artificial muscles becomes crucial. Since external attachment devices greatly hinder the miniaturization and use of artificial muscles, we propose a light-driven approach to get rid of these limitations. In this study, we report a miniaturized fiber-reinforced artificial muscle based on mold editing, capable of bending and axial elongation using gas-liquid conversion in visible light. The minimum volume of the artificial muscle prepared using this method was 15.7 mm³ (d = 2 mm, l = 5 mm), which was smaller than those of other fiber-reinforced pneumatic actuators. This research can promote the development of non-tethered pneumatic actuators for rescue and exploration, and create the possibility of miniaturization of actuators.

Artificial muscles can use low-complexity structures to crawl, swim, and grasp irregularly shaped objects[1-3]. These actuators can bend, contract, or elongate depending on various control methods[4,5], including pneumatic[6-8], electrical[5,9,10] or magnetic[11,12]. Responsive functional materials, such as liquid crystal polymers[13-17], hydrogels[18-20], electroactive polymers[21], and shape-memory alloys, are employed in these muscle-like actuators[22]. Although the chemical, electrical, and magnetic activation of soft actuator robots has been demonstrated[23], pneumatic robots powered by pressurized air have drawn considerable attention owing to their simple and rapid actuation[24,25]. The widely used pneumatic artificial muscle comprises pumps, an elastomeric tube, a braided sleeve, and two-end fittings[26-28], with one-end fittings incorporating a fixed input channel for liquid or gas. When the gas or liquid enters the channel, a drastic pressure change induces a fast response and effective actuation. The elastomeric tube is fitted into the braided sleeve, and upon expansion the muscles contract or extend depending on the resting braid angle[29-31]. Although pneumatic devices provide greater output and faster response times, remote-control and miniaturization applications remain challenging owing to

the limitations of valves, pumps and connecting pipes[32]. We note that some work has been done on the thin McKibben[33,34], but due to the limitations of the energy device, it is difficult to further reduce the size of the actuator. At the same time, the exploration of miniaturized pneumatic muscles has been impeded by the limitations of control valves, limiting the size and further applications of wireless drivers[35-37].

To address these challenges, the liquid-gas phase transition can be used as a pneumatic working medium as an effective alternative to conventional electromechanical devices[38,39], enabling wireless control by external fields, such as magnetic and light stimuli[40,41]. These materials rely on the mechanical force produced by the rapid expansion occurring at the phase-transition temperature. Recent devices have introduced liquid phase-change materials within balloons or thin films, creating enlarged cavities that undergo liquid-gas phase changes via electrical or magnetic heat[42-46]. This enhancement removes the need for external connections to power resources, batteries, and connecting pipes and pumps. Nowadays, the use of phase change liquids for wireless gas drive has become the choice of more researchers. In order to obtain greater deformation, researchers often choose rubber with

[1]Key Laboratory of Bio-inspired Materials and Interfacial Science, Technical Institute of Physics and Chemistry, CAS, Beijing 100190, P. R. China. [2]School of Future Technology, University of Chinese Academy of Sciences, Beijing 100049, P. R. China. [3]Binzhou Institute of Technology, Weiqiao-UCAS Science and Technology Park, Binzhou City, Shandong Province 256606, China. [4]These authors contributed equally: Wenfei Ai, Kai Hou. ✉e-mail: longyue@mail.ipc.ac.cn; songkai@mail.ipc.ac.cn

high permeability to low boiling point liquids as a carrier, which will lead to liquid leakage. Therefore, the sustainable use of devices will become an issue.

Admittedly, there has been a lot of work in the field of optically controlled small drives[47–50], but most of them can only rely on asymmetric forces to perform relatively simple bending deformations. Here we show a work that not only combines light control and pneumatics, but also miniaturizes pneumatic artificial muscles. On this basis, we also added three application scenarios to show the diversity of its functions. Our study proposes a miniature artificial muscle wirelessly for sustainable use controlled by light. We wrapped the outer layer of the device with a PTFE(Polytetrafluoroethylene) fiber membrane, depending on the fiber orientation, to limit the radial expansion of the device to achieve maximum axial elongation and bending. This design enables the artificial muscle to serve as functional units for robotics. In contrast to the conventional method of creating cavities (injection followed by sealing), we poured a prepolymer containing carbon and a low boiling liquid into a transparent cylindrical mold and cured it under ultraviolet light. Carbon blocks UV rays from penetrating and prevents the interior of the elastomer from curing, resulting in a cylindrical cavity containing the prepolymer fluid. This method prevents leakage during fabrication and improves the sealing properties of the elastomer. Consequently, the artificial muscle achieves enhanced tightness, reducing the diameter and length of the artificial muscle to 2 mm × 5 mm, respectively. After the liquid inside the artificial muscle is consumed, we can soak the artificial muscle in a low boiling point liquid to preserve and restore it to drive performance. Under light irradiation, the artificial muscle exhibits efficient expansion, elongation, and programmable bending. These properties help to fabricate devices for object grasping or crawling in narrow channels.

## Results

### Material composition and mechanism

First, we designed a simple and universal manufacturing method for assembling cylindrical elastomers based on mold editing. We used mechanical agitation as a means of dispersing low boiling point liquids with a rotational speed of 235 rpm. As shown in Fig. 1a, we cured a prepolymer mixed with a kind of polyurethane acrylate named H810, nanometer-sized carbon powder, and a low boiling liquid (1,1,1,4,4,4,-Hexafluorobutene) under UV light. The carbon nanoparticles obstructed UV light absorption, preventing the prepolymer mixture in the actuator from curing and remaining in the fluid state. As shown in Fig. S1, the PTFE's optic images of the fluid inside the actuator demonstrate that the low boiling liquid is present as droplets, with an average diameter of 2.9 μm. These droplets vaporized on heating the actuator, increasing the internal pressure and further leading to the expansion of actuator.

We employed a PTFE film with a single-oriented fiber wrapped on its surface as a limiting layer in the direction of a vertical cylinder to achieve unidirectional contraction and expansion resembling a biological muscle fiber. In Fig. S10, the optical images of PTFE film before and after stretching were shown. The PTFE film readily fell off the elastomer's surface without an adhesion layer after wrapping. Therefore, we applied and cured the prepolymer onto the film to ensure the PTFE film remained securely in place. Figure 1b illustrates the phase transition of a low boiling liquid. The complete deformation process of the actuator is shown in Fig. 1e and Supplementary Movie 1. When the PTFE film fibers were perpendicular to the elastomer's axis, the actuator extended in a single direction (axial) upon heating and returned to its initial length after cooling. Figure 1c shows a schematic diagram of the changes in the fiber membrane during actuator

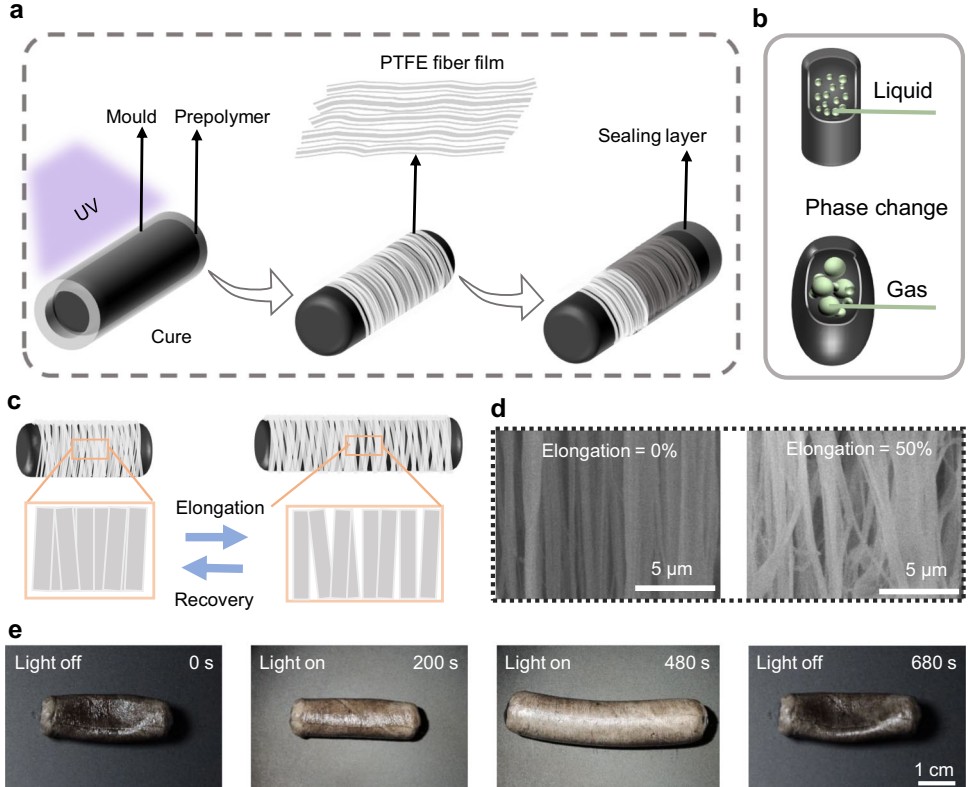

**Fig. 1 | Preparation and deformation mechanism of artificial muscle.**
**a** Schematic diagram of artificial muscle preparation. **b** Phase transition of low boiling liquid. **c** A schematic illustration of the deformation of an artificial muscle. The enlarged image shows the changes in the space between fibers as the membrane is stretched. **d** SEM images of fiber membrane under 0% and 40% elongation. **e** Complete deformation and recovery of artificial muscle. PTFE is the abbreviation of polytetrafluoroethylene.

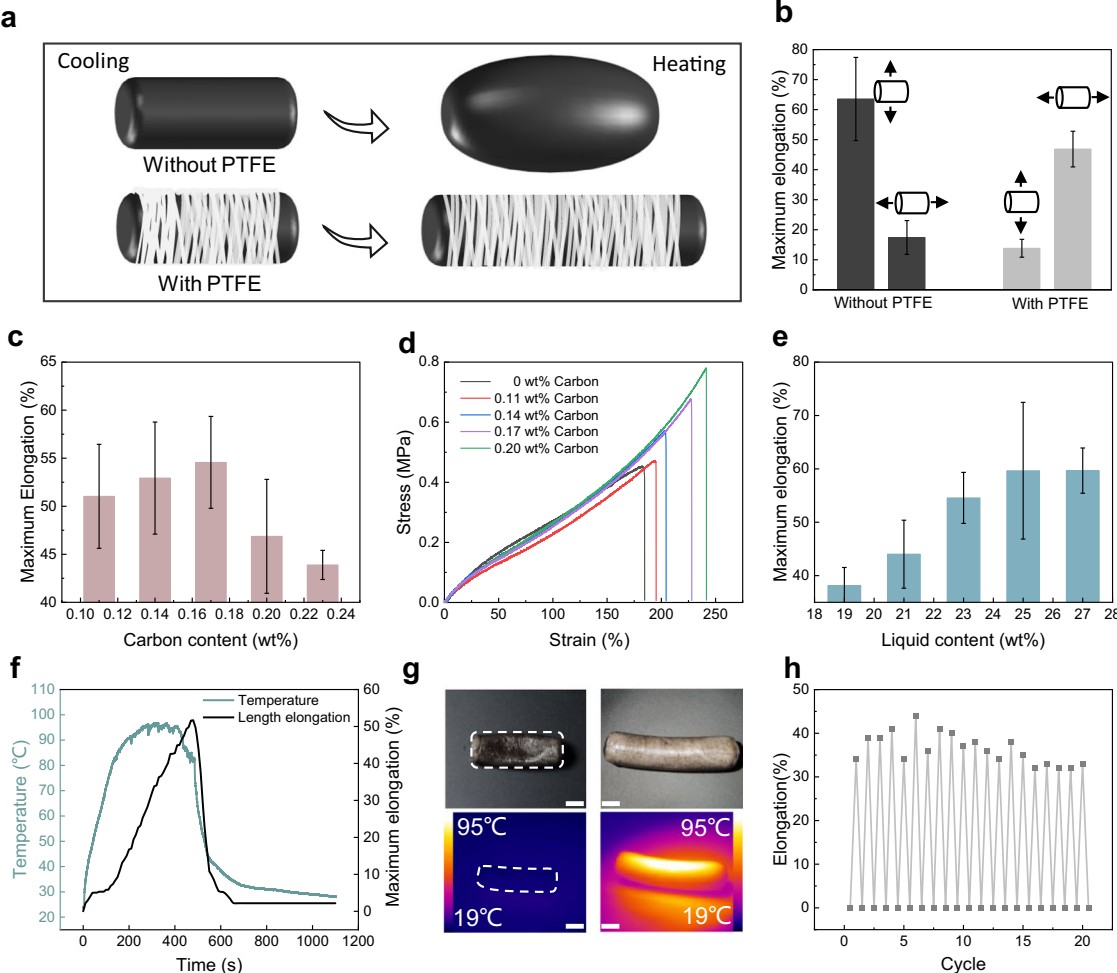

**Fig. 2 | Characterizations of the artificial muscle. a**, **b** Maximum axial and radial elongation of artificial muscle vs. elastomer (depending on whether it is wrapped in PTFE). **c** Maximum elongation of artificial muscle with different carbon content. **d** Stress−strain curves of different carbon content. **e** The maximum elongation of different liquid contents (The diameter of the sample is 9 mm, and the length is 30 mm) **f** Temperature and elongation changing over time. (The light power density is 100 mW/cm². **g** Infrared image of artificial muscle after original length deformation (room temperature is 19 °C, scale bar is 1 cm). **h** Eight cycles of artificial muscle deformation at room temperature. The light power density is 100 mW/cm² in (**c**, **e**, **f**, **g**) and 120 mW/cm² in (**h**). All the error bars correspond to s.d. (n = 3).

extension. Figure 1d shows SEM images of the fiber membrane at 0% and 50% elongation, indicating that the space between the fibers increased with stretching. Based on the photothermal effect, carbon nanoparticles heat the actuator in a photoenergy-thermal conversion system. Consequently, the actuator heats up quickly under light, whereas the liquid inside easily vaporizes because of its low boiling point (33 °C), leading to a drastic change in volume and pressure, enabling axial extension of the artificial muscle within the constraint of the fiber film. When the light source is removed, the gas returns to a liquid state, initiating muscle contraction. Additionally, the maximum axial and radial elongations of the artificial muscle and ordinary cylindrical elastomers were compared to demonstrate the radial constraint of the PTFE membrane on the elastomers. Previous studies have demonstrated the expansion of cylindrical elastomers under pressure. Figure 2a shows that the radial elongation of the elastomer without the PTFE coating is greater than the axial elongation[51], whereas that of the PTFE-coated elastomer is lower, confirming that the PTFE membrane restricts radial expansion and facilitates axial elongation of the actuator. In Fig. 2b, the axial and radial elongation of a PTFE wound and a non-PTFE wound drive were compared. We can see from the image that the radial elongation of the drive varies by up to 63% before winding, while the axial elongation is only 17%. After wrapping the PTFE film, the radial elongation of the drive is reduced to 13% and the axial

elongation is increased to 47%. Therefore, we can conclude that the winding of the fiber reduces the radial elongation by 50% and increases the axial elongation by 30%. In this way, we succeeded in changing the main direction of the deformation from radial to axial. Therefore, our winding method effectively increases the structural constraints in a single direction to transform the original radial expansion into axial elongation, which is similar to the deformation mechanism of fiber reinforcement.

## Mechanical properties and optimization

We studied the effects of the nanocarbon powder and low boiling liquid contents on the artificial muscle's tensile strain to optimize the performance of artificial muscles. Figure 2c shows that when the liquid content is fixed at 23 wt%, and the carbon content is limited within the range of 0.11−0.17 wt%, the maximum axial elongation of the artificial muscle gradually increases, reaching a maximum of 54.6%. This behavior can be explained using the principle of particle strengthening[52,53]. The thickness of elastomer films with different carbon content using a thickness gauge was compared in Fig. S2. The film is not uniform, about 100 microns, which may be caused by the lighting mode of the handheld UV lamp, and improving the lighting mode such as ring UV curing may be beneficial to the uniformity of the film thickness. However, the uniformity of film thickness has no

obvious effect on the overall performance of the driver. Figure 2d shows that with increasing carbon content, the theoretical maximum strain of the elastomer film also increases gradually. However, as shown in Fig. 2c, when the carbon content of the artificial muscle was greater than 0.17 wt%, the increase in particles decreased the air-tightness of the elastomer membrane, reducing the maximum elongation. It can be further explained from two aspects. First, when the outer layer is photocured, the doped carbon powder, as a solid particle, does not participate in the chemical crosslinking in the pre-polymerization solution. Therefore, excessive carbon powder will cause the structure of the polymerized elastomer to be loose and the air tightness to be weakened. Secondly, the absorption of ultraviolet light by the carbon powder will affect the thickness of the polymer film of the driver, and the increase of the carbon content will reduce the thickness of the polymer film, which also leads to the reduction of the air tightness of the driver to a certain extent. Therefore, 0.17 wt% was chose as the optimal value. The increase in fluid content significantly increases the volume of the artificial muscle after expansion. Consequently, we recorded the maximum axial elongation of the artificial maximum elongation of the artificial muscle up to 60% was observed when the fluid content was 25 wt% or less, attributed to an increase in the liquid content. However, we observed that at 25 wt% liquid content, the artificial muscle was damaged after reaching the maximum elongation. Consequently, we selected an artificial muscle with a 23 wt% liquid content for testing. As the liquid content increased, the maximum elongation did not change significantly because the excessive gas broke the artificial muscle, resulting in a loss of deformability. Figure S3 illustrates the influence of liquid mass fraction on the strain of the elastomer film. To explore the relationship between elongation and temperature, we recorded the temperature and elongation changes of the artificial muscle upon light exposure. Owing to the photo-thermal effect of carbon, the artificial muscle heats under light irradiation, and the liquid droplets gradually vaporize during heat transfer. Consequently, the muscle elongation gradually increased with increasing temperature. When the average surface temperature was 50 °C, the actuator began to elongate and reach an equilibrium in 460 s (Fig. 2f). Subsequently, the light was turned off, and the photo-thermal effect disappeared. Additionally, the actuator length gradually decreased with decreasing temperature. As shown in Fig. 2g illustrates the variation in artificial muscle length with increasing temperature. The maximum deformation response of each sample was recorded eight consecutive times under light. As shown in Fig. 2h, the sample with a liquid content of 19 wt% can complete 20 reversible elongations within 68 min, and the average elongation was maintained at 36.4%. The optical power density of xenon lamp in the experiment is 120 mW/cm². It is found that with the increase of the number of cycles, the elongation will decrease. Therefore, gas leakage was considered as a potential cause of elongation loss. First, the pores on the specimen surface expand after elongation, and the high temperature at this time speeds up the gas seepage rate. Secondly, due to the addition of nano carbon powder when curing the elastomer, the entire elastomer is not crosslinked tightly, and the structure becomes relatively loose after repeated stretching, and it cannot maintain the initial state. In order to reduce the influence of gas leakage on the elongation, the sample with damaged elongation was soaked in a low boiling point liquid for supplementation. Through the elongation verification of Fig. S9, we show that this method is sufficient to restore the specimen to the original driving effect as long as it is soaked for a sufficient time.

We measured the maximum unidirectional force obtained at various blocking strain levels in the range of 0–50% to investigate the blocking orientation force characteristics of the actuator during photoheating. The actuator was fixed in a cylindrical container to eliminate radial volume expansion during this process. The maximum unidirectional force for an unstrained sample weighing -2.4 g was -6.3 N, Fig. S4). We understand reviewers' concerns about output force

application scenarios. In order to better show the application prospect of the output force, we designed a light-controlled thruster in Fig. S7 and Supplementary Movie 6. We made a driver with $L = 3.6$ cm and $d = 9$ mm, fixed its left side and placed a 100 g weight on the right side. Under light irradiation, the driver shifted the weight to the right side by 2.1 cm. The maximum static friction force between the weight and the surface of glass has been measured to be 0.2 N.

## Applications in robotics

In this study, we have demonstrated the use of artificial muscles as actuators in various robotic applications. We successfully reduced the size of the artificial muscle to the millimeter level by taking advantage of integrated mold preparation and realized a size leap to explore the potential of artificial muscles in miniaturization applications. As shown in Fig. 3a, b, we fabricated a miniature artificial muscle walking device composed of an artificial muscle and a transparent exoskeleton that could be driven by light. The minimum volume of the artificial muscle prepared using this method was 15.7 mm³ $(d = 2$ mm, $l = 5$ mm). The device moves forward to the right by extending and retracting, depending on the position of the artificial muscle and the direction of the forces. In supplementary information, we present a force analysis of the device during extension and retraction. Figure 3c demonstrates the complete unidirectional walking process of the miniature device. When the device is extended and contracted under a light switch, the asymmetric friction force on both sides causes it to move forward to the right. Under visible light exposure, the artificial muscle extends axially, generating asymmetric thrust that propels the device to unfold with greater displacement to the right. The artificial muscle contracted when the light was turned off, resulting in greater displacement to the left. Simultaneously, to increase the friction on the right side, we created multiple sharp angle shapes (Fig. S6) on the part that contacted the ground. The continuous walking process is illustrated in Fig. 3d and Supplementary Movie 2, where the miniature walking device advanced by 3 mm after four cycles of light switching.

To demonstrate the movement function of artificial muscles in complex environments, such as tubes, we studied and manufactured composite artificial muscle devices bioinspired by earthworms. Without any external power supply or pressure pump, our self-containing artificial muscles could wriggle through a tube under a mobile light source. We further show the three functional structures of the device: the fore and back feet ($d = 9$ mm), comprising hemispherical elastomers that expand freely as anchors, and the body ($d = 9$ mm and $l = 34.8$ mm), comprising an artificial muscle as the moving component (Fig. 4a). Figure 4a, b and Supplementary Movie 3 illustrate the complete process of the composite artificial muscle device crawling through a tube with an internal diameter of 14 mm. Initially, the light spot irradiates the backfoot, and under the photothermal effect, the liquid inside the backfoot vaporizes, causing it to expand and squeeze the pipe, establishing the anchoring (Fig. 4b, I). Subsequently, as the light spot gradually covers the actuator body, the device undergoes forward movement by expanding and extending (Fig. 4b, II). The backfoot remained in the light and retained its anchored position until the elongation of the body reached its limit (Fig. 4b, III). The light spot then gradually leaves the backfoot and moves to the forefoot, gradually anchoring while the body maintains its ultimate elongation until the forefoot is anchored (Fig. 4b, IV). Subsequently, as the light spots leave, the body begins to indent in the forward foot direction (Fig. 4b, V). Finally, when the illumination ended, the anchoring of the forefoot ceased (Fig. 4b, VI), completing the in-tube displacement process of the composite artificial muscle device. The composite artificial muscle device achieved multiple continuous displacements in the tube, with a total travel distance of 5 cm, owing to the repeatability of the braking of our artificial muscle. Figure 4c shows the distance moved between the front and back points of the body over time during a single displacement. The foot anchoring causes deviations in the beginning and

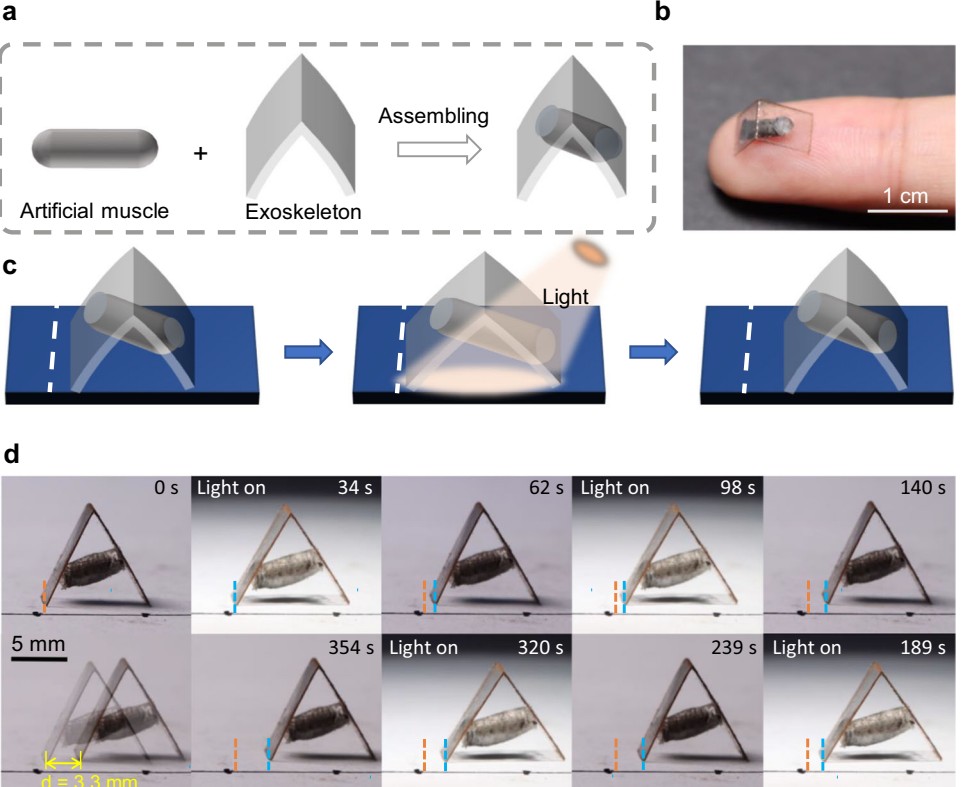

**Fig. 3 | Miniature artificial muscle walking device. a, b** The assembly process of a miniature artificial muscle driver and its optical image. **c** A diagram of the miniature device crawling on a flat surface. **d** The device crawls in four steps on the plane, with a total advance of 3.3 mm. (The light power density is 150 mW/cm².)

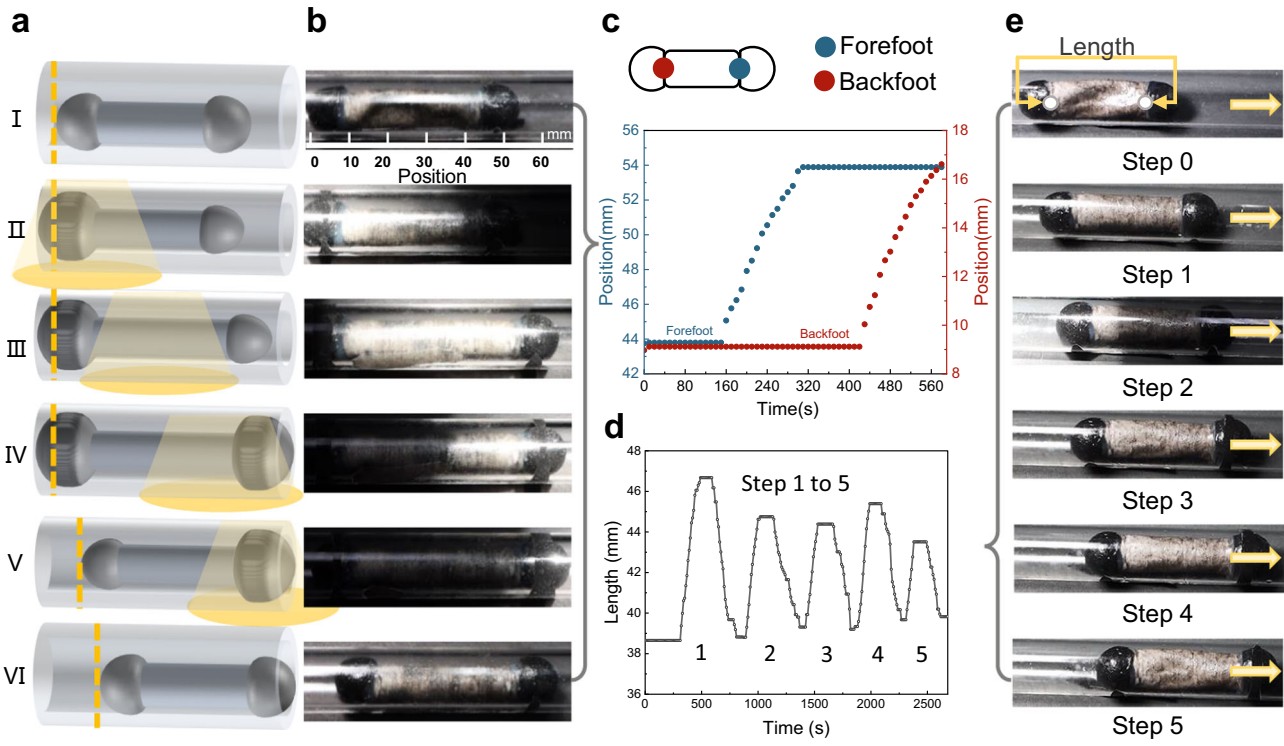

**Fig. 4 | Actuation performances of the earthworm-like compound artificial muscle device in the pipe. a, b** The crawling diagram and optical image of the earthworm-like compound artificial muscle device in the pipe. (The inner diameter of the tube is 14 mm). **c** Changes in the position of the front and rear points of the device over time. **d** The cycle of the body length changes under UV light. **e** Five consecutive movements of the equipment in the tube. (The inner diameter of the tube is 14 mm and the light power density is 150 mW/cm².)

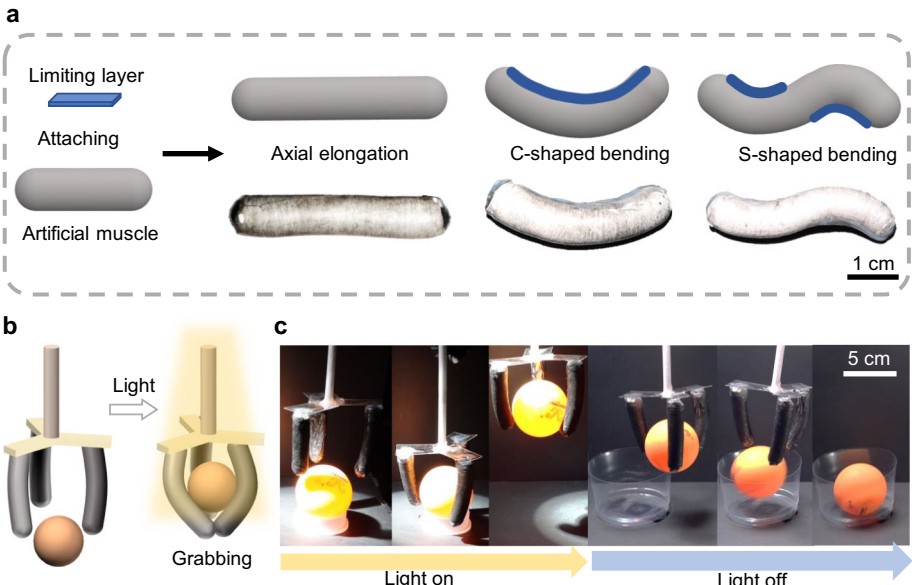

**Fig. 5 | The functional demonstration of a gripper made of artificial muscle.** **a** Design of flexural deformation of artificial muscle. **b** Gripper design diagram. **c** The grasping and releasing process of the device under light illumination. The light power density is 150 mW/cm$^2$.

end times of the body's front and back, enabling unidirectional movement of the entire body. Figure 4d and Supplementary Movie 4 shows the changes in the length of the actuator body during multiple displacements, demonstrating the capability of a composite artificial muscle device to achieve multiple displacements through continuous expansion and contraction of its body muscles. The origin of the coordinate system was set at the vertex of the posterior hemisphere. It should be noted that in the pipeline crawling experiment we used a light mode of delayed segmenting irradiation, and the displacement generated during the anchoring may be lost due to the delay of the optical operation. After the anchoring of the front foot is completed, the illumination of the front foot is prematurely cancelled during the waiting time for the body to retract, so that the anchoring is cancelled when the body is not fully recovered. Therefore, the body loses its anchor point and begins to retract toward the center, driving the front foot to retreat, resulting in the loss of displacement. Secondly, the friction in the pipeline will also have a certain impact on the displacement.

Based on the aforementioned uniaxial extension artificial muscle, we studied and designed a flexible artificial muscle (Fig. 5a) to demonstrate various applications of artificial muscles. The fiber membrane lost its ability after incorporating the limiting layer to extend along the cylindrical axis. When heated by light, a flexible artificial muscle relies on the strain difference introduced by the elastic layer to achieve a C-bend. Similarly, we used a limiting layer on the top and bottom of the cylindrical artificial muscle close to the two-end fittings to achieve an S-shaped bend under UV light.

Furthermore, the controllable bending capability of artificial muscles demonstrates their value in grasping objects. As shown in Fig. 5b, c and Supplementary Movie 5, we assembled a gripper comprising three C-shaped muscles 8 cm in length to grasp the ping-pong ball under light with air heat flow and released the ball when the light ended.

## Discussion

In the statistical analysis of the maximum elongation of artificial muscles, we noticed significant instability in the data and analyzed the possible potential factors contributing to this phenomenon. Since the PTFE film itself does not have adhesion, the fixation of the film

becomes a key factor limiting the radial expansion of the driver. After wrapping the PTFE film, we applied a prepolymer solution to its outer layer and cured it as a holding measure for the film. However, only relying on the outer layer fixation cannot make the driver obtain a more efficient radial binding force, that is, during the drive process, the stress at the interface of the PTFE film will exceed the bearing capacity of the fixed layer, resulting in a rapid release of stress in a small range, resulting in a loss of elongation or damage to the device. To further improve the stability of the artificial muscle, we tried to apply a prepolymer solution to the PTFE overlapping interface and cure it, using it as an adhesive to bind the PTFE membrane interface. This may cause uneven force on both sides of the drive, making it bend, but it is conducive to stability. Regarding to any possible leakage issues with the drive, the comparison between the elongation of the sample in the initial state and that after 4 h soak in the liquid is shown in Fig. S9.

For further large-scale manufacturing, especially for the mass production of miniaturized drives, we can first prepare long and thin elastomers to wrap around the whole, and then cut different lengths according to the need. Finally, the intercepted miniaturized actuators can be quickly sealed by the photocuring properties of the polymer.

In summary, our miniaturized fiber-reinforced artificial muscles can perform tasks remotely under light, and their diverse designs (elongation and bending) provide opportunities for complex integrated mechanisms. Consequently, our artificial muscles could use different structural designs to move continuously through flat ground and pipes. Additionally, our artificial muscles are smaller than those of other fiber-bound pneumatic actuators. In the future, we can further reduce the size by microcapsules, electrospinning and colloidal assembly to expand the range of practical applications in biology, medicine, and other fields.

## Methods
### Fabrication of the artificial muscle
We used H810 adhesive (viscosity 30,000 cps.25 °C, hardness 32 A) as the elastomeric matrix material, purchased by Suzhou Jing Cheng Technology Co., Ltd., China. 1,1,1,4,4,4,-Hexafluorobutene, as the active phase change material, was purchased by Shanghai Rui Yi Environmental Protection Technology Co., Ltd. Nanocarbon powder

was used as a light absorption and photothermal conversion material. Different mass fractions of nanometer-sized carbon powder and 1,1,1,4,4,4,-Hexafluorobutene with H810 were thoroughly mixed for ~30 min to prepare the material, which was then introduced into a mold and solidified. After stripping, the cylindrical surface was coated using a polytetrafluoroethylene film. Finally, the outermost layer was coated using a mixed material for solidification.

### Fabrication of the miniature artificial muscle walking device
The exoskeleton of the miniature artificial muscle was cut using a laser to a length of 20 mm and width of 3 mm before bending. The miniature artificial muscle with cylindrical mold of 2 mm × 5 mm. The assembly of the exoskeleton and the miniature artificial muscle was achieved through the curing and bonding of H810.

### Fabrication of the earthworm-like compound artificial muscle device
The pipeline-crawling composite artificial muscle device used a hemispherical mold ($d = 9$ mm) to create the foot. Xenon lamp (SHX-F300, Beijing NBET Technology, Beijing, China) was used for light heating.

### Fabrication of the gripper
H810 was solidified on the sides of each of the three artificial muscles as a limiting layer, and then were glued to the laser-cut supporting bone to complete the gripper assembly. The adhesive is H810.

### Characterizations
A digital camera (Canon EOS 90D) was used to record changes in artificial muscle elongation and perform statistical calculations using ImageJ. A universal testing machine (Instron3365, INSTRON, USA) was used to record the temperature changes under light exposure. A nano-search microscope (OLS4500, Japan) was used to observe the low boiling droplets in the prepolymer. The stress–strain curves were obtained on an Instron 5566 universal tester (Instron, Norwood, MA) at a tensile rate of 20 mm min$^{-1}$. Each sample ($5 \times 1$ cm$^2$) was tested four times. For the blocking force experiment, a cylindrical artificial muscle sample ($l = 30$ mm, $d = 9$ mm) was placed in a polymethyl methacrylate (PMMA) cylinder, fixed to the bottom and connected to a force gauge at the top (Series5 M5-2, MARK-10, USA). The experimental setup is shown in the Fig. S4. A sample of a constant size was placed in the cylinder, the position of the inductor was changed, and the force was recorded. Each test was conducted with three samples.

## Data availability
All data needed to evaluate the conclusions in the paper are present in the manuscript and Supplementary Information.

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

## Acknowledgements
W.A., K.H., J.W., Y.L., and K.S. were supported by the Ministry of Science and Technology of China (No. 2019YFA0709200 and 2022YFA1503000); the National Natural Science Foundation (21988102); and TIPC Director's Fund.

## Author contributions
W.A., K.H., and Y.L. conceived the study and designed the experiments. K.S. supervised the project. W.A. and J.W. performed the experiments and characterization. W.A., K.H., Y.L., and K.S. wrote the paper. All the authors have discussed the results and revised the paper for submission.

## Competing interests
The authors declare no competing interests.
