## [Peer Review File · Nature Communications]

REVIEWER COMMENTS

Reviewer #1 (Remarks to the Author):

This study presents a miniaturized, untethered McKibben muscle for soft robots, driven by photothermal-induced liquid-gas transformation. The incorporation of carbon particles into the silicone rubber material facilitates a self-shielding effect, enabling spontaneous core-shell formation under UV light. This streamlines manufacturing and reduces actuator size. By adding a restricting layer of PTFE fiber membrane, the radial expansion of the artificial muscle is effectively suppressed, thereby achieving anisotropic driving ability. After the material and performance characterization, three untethered McKibben muscle functions are demonstrated: a climber on flat plate, a worm-inspired soft robot wriggling in tube, and a gripper consisted of three independent artificial muscles. The untethered artificial muscle presented in this work has potential in enhancing high-force soft robotic applications, with its unique fabrication method enabling downsizing of liquid-gas phase transition-driven actuators.

However, the novelty, completeness and importance of the current submission falls short of the standards of Nature Communications. Addressing and clarifying the following aspects will be crucial before considering publication elsewhere:

The soft actuator introduced in this paper relies exclusively on a low-boiling-point liquid (25°C) as its liquid-gas phase change working medium. However, the actuator does not demonstrate any advantages in terms of operational capabilities, driving performance, size, or structural complexity when compared to previous publications focused on liquid-gas transmission actuators. Furthermore, it's worth noting that the manual winding of constrained fibers contributes to the intricacy of the manufacturing process, thereby limiting the potential for achieving high-precision production and large-scale assembly. This limitation is evident from the prevalent use of sample sizes at the centimeter scale. More efficient designs for anisotropic elongation and force output have already been presented ("Wireless soft actuator based on liquid-gas phase transition controlled by millimeter-wave irradiation." IEEE Robotics and Automation Letters, 2020).

Simultaneously, the manuscript neglects to provide information regarding the energy density of the high-power light source, giving rise to concerns about the actuation efficiency. Another notable concern pertains to the stability of the driver's performance, stemming from the utilization of a working fluid with a boiling point closely aligned to room temperature. This decision introduces considerable uncertainties regarding performance reliability, and no obvious advantage has shown considering the high temperature applied to stimulate the actuator. While the proposed fabrication method showcases intriguing elements, its genuine potential remains unexplored due to the absence of comprehensive analysis, theoretical foundations, and adequate characterization.

1. The manuscript emphasizes "light-driven" yet lacks critical light power density details throughout. To ensure accuracy and readability, please provide comprehensive disclosure of experimental processes and results in the main manuscript and supplementary information.

2. In light of the promising driving performance achieved by the proposed soft actuator, a more comprehensive comparison with relevant reported works in the field is needed. For instance, Tang's work (Wireless miniature magnetic phase-change soft actuators, *Adv. Mater.*, 2022) demonstrates a substantial output force and working capacity, Kurumaya's work (Design of thin McKibben muscle and multifilament structure, *Sensors and Actuators A: Physical*, 2017) shows a pneumatic McKibben muscle design with comparable size, as well as Cacucciolo's work (Electrically-Driven Soft Fluidic Actuators Combining Stretchable Pumps With Thin McKibben Muscles, *Frontiers in Robotics and AI*, 2020). A thoughtful discussion that contextualizes the proposed soft actuator's performance against such benchmarks will provide valuable insights into its potential and unique attributes.

3. The manuscript intriguingly explores a low-boiling point liquid-based actuator requiring a relatively high driving temperature. I'm confused by the mismatching in temperatures of the liquid boiling point and the heating process, and encourage a thorough investigation into the interplay between temperature, inner pressure, and elongation. Such an exploration could shed light on the underlying physics and mechanisms governing this behavior, potentially leading to insights for optimizing the actuation process.

4. The manuscript reveals an actuation process that, while promising, exhibits notable inefficiencies in terms of duration and speed (8 minutes for one cycle of driving and 40 minutes for achieving a moving distance of 50% of the body length). To advance the practical utility of the proposed actuator, I recommend delving into potential structural design improvements that could enhance actuation efficiency. This could involve exploring innovative configurations or materials that facilitate faster and more effective actuation.

5. The self-shielding effect of UV light demonstrated in the fabrication process is noteworthy for simplifying manufacturing, and endows the possibility of more complex 3D shaped actuators with one-step fabrication. However, the manual wrapping of the PTFE fiber membrane (Line 63) appears labor-intensive. As a suggestion, the authors might consider optimizing this step or exploring alternative fabrication strategies that further emphasize the advancement in manufacturing techniques.

6. Demonstrations highlighting the actuator's capacity for generating substantial output force would be invaluable in showcasing its potential applications. This could include scenarios where the actuator's

force output is particularly advantageous, underscoring its practical utility and setting the stage for broader adoption.

7. The manuscript introduces a significant advancement in actuator downsizing. To underscore this accomplishment, I suggest contemplating the feasibility of further miniaturizing the actuator to submillimeter dimensions. Such exploration could underscore the uniqueness of the proposed fabrication method compared to existing works and highlight its potential for even smaller-scale applications.

8. How did the shell thickness shown in Fig. S2 measured? Whether the thickness varies between different fabrication batches?

9. Long-term stability remains a critical aspect for practical deployment. especially when the phase change liquid holds a boiling point of 25°C. The internally depressed sample in Fig. 2G indicates the severe leak of contained low-boiling point liquid during the fabrication and driving process. Also, 8 times elongation test is inefficient to prove the high reversibility (Line 147). How long does the actuator is expected to be working properly, could the authors comment on this?

10. In the introduction section (Lines 51-53), I suggest redirecting your focus from pneumatic devices to the domain of remotely controlled miniaturized soft robots/actuators. This adjustment is recommended considering the extensive body of existing literature in this field. It is advisable to avoid delving into an extensive and redundant historical account of pneumatic McKibben muscles, as this could potentially dilute the engagement of both your time and the readers' attention. Furthermore, in Lines 55-62, you aptly outline the developmental trajectory of liquid-gas phase transition-based soft actuators. However, the introduction would benefit from the inclusion of pressing and contemporary challenges currently encountered within this field. Presenting these challenges succinctly and directly will enable a more engaging introduction, allowing readers to grasp the distinctive contribution of your work and its targeted approach to addressing a specific problem.

11. The data plots in Fig. 2C,E reveal a notable instability in the elongation of the actuator driven by phase change, characterized by a considerable variation range. It would greatly enhance the manuscript's depth to delve into the potential factors contributing to this observed phenomenon. Furthermore, an insightful discussion of possible strategies or modifications that could be implemented to mitigate this instability would provide readers with valuable insights into enhancing the actuator's consistency and reliability.

12. In Lines 125-127, the conclusion drawn regarding the relationship between particle increase and elastomer membrane airtightness, as indicated in Fig. 2C, warrants further elucidation. The current presentation of the data may not fully substantiate this connection.

13. Detailed discussion is needed in Discussion Section.

14. What does “when the container expands indefinitely” mean (Line 128)?

15. What does “the elongation began to increase gradually, reaching a maximum of ~460s” mean (Line 142)?

16. Ensuring consistent abbreviation usage is pivotal for reader comprehension. In Line 64, it is advisable to introduce the full name before employing an abbreviation to enhance clarity upon its initial appearance.

17. Please mark the liquid droplet in Fig. S1, it's hard to identify useful information from the figure.

18. Also, what kind of mixing strategy was employed to make the droplet distributed uniform in the prepolymer with a diameter of around 3 μm ? The disclosure of necessary experimental steps is essential.

19. Please provide the data of applied temperature in Fig. 2C (Line 120) to bolster data transparency.

20. To enhance the manuscript's organization, providing explicit references to specific figures within the supplementary information using designations like "Fig. S4" instead of the more general "Supplementary figure" would enhance navigability for readers. (Line 268).

21. When characterizing the elongation of the actuator, discussing any observed changes in the actuator's radius concurrently would provide a more comprehensive understanding of how these properties correlate.

22. Could the author comment on how to improve the fabrication process to meet the needs for massive production?

23. How does the UV light density and the operating time influence the thickness of the shell? Will the uncured prepolymer gradually cure during the actuation process or after a long time?

24. A comprehensive rewrite of the abstract is recommended to concisely emphasize the unique contributions and novelty of this work. Furthermore, a meticulous language polish throughout the manuscript will ensure clarity and precision, enhancing the presentation of your findings.

Reviewer #2 (Remarks to the Author):

This manuscript presents a novel approach by proposing a miniaturized, untethered McKibben Muscle powered by an internal phase change material, which is activated by UV light. While the study provides valuable insights, I have several key comments that need addressing.

1. Relevance of UV Activation: The primary appeal of the work lies in its attempt to address the challenge with bulky pumps in pneumatic actuators – a fact I acknowledge and appreciate. However, I question the practicality of using UV light as a driving mechanism. The abstract suggests applications in "miniaturized medical equipment, rescue, and exploration", but it remains unclear how UV light would be suitable or practical for all these contexts. Could the authors illustrate with a specific scenario? While the manuscript emphasizes the off-board energy source, it's pertinent to remember that this energy still needs to be supplied for the actuator to function. It means the lamp has to be around. Additionally, the penetration depth of UV light in various practical situations should be discussed.

2. Light Intensity Clarification: The manuscript seems to have overlooked the specifics regarding the intensity of the UV light used. While it mentions the employment of a Xenon lamp, there's no elaboration on its intensity. It's important to understand if the impressive performance of the actuator is primarily due to high light intensity or the inherent design of the actuator. This distinction is crucial as many locomotion demonstrations seem to be driven by the light's direction and intensity rather than the actuator's intrinsic properties.

3. Comparison with Prior Work: While the paper makes reference to existing literature (line 60), I believe a more comprehensive review and comparison are warranted. More crucially, the manuscript should elucidate why its approach and results are superior or distinct from previously published studies, especially in those application scenarios mentioned.

In conclusion, while the manuscript offers an intriguing concept, it would greatly benefit from addressing the aforementioned concerns. There's a plethora of work in the domain of light-controlled miniature robots, and this study should firmly establish its unique contribution and relevance in the field.

Reviewer #3 (Remarks to the Author):

Overall, this is an interesting and reasonably well written paper. The proposed concept for a self-contained, light driven fluidic muscle actuator appears to be effective and simple. The description of the concept and the associated methodology is fairly well presented, and the different demonstrations showing the working of the concept are insightful. Some of the details of the technical work are a bit weaker though, and some issues present in the results are not adequately discussed. In particular, the significant loss in performance seen in initial cycling with both the muscles themselves and the tube crawler device need further discussion. Ideally a more thorough running in would have been performed to see if the loss in deformation saturates to a consistent level, after which the data for presenting should have been taken. It would have been particularly useful to see the force versus displacement behaviour characterized to better understand the ability of these devices to do work. The bending walker and tube crawler are only carrying their self-weight, and since force/energy capability aren't measured, we can't make any conclusions as to their ability to do meaningful work in the real world. A single force number is presented, but it is completely unclear to this reviewer why they chose to put the muscle into a separate external tube (thereby artificially and unrealistically constraining its diameter) when taking this measurement, and as a result this number doesn't seem to mean anything. Ultimately though, the work is quite interesting and if these minor weaknesses can be corrected then I believe it is worthy of publication. Some detailed comments follow:

Pg. 3 – “classic structures of McKibben muscles develop” - “develop” is misspelled and not needed

Pg. 4 – The moulding technique and partial curing to create an outer membrane and an internal working fluid is clever, but I am wondering about the long term durability – in particular the likelihood of continued curing over time and a slowly thickening wall.

Pg. 7 – The quoted maximum elongation of 47.07% after 8 cycles is not a fully accurate description, as the actuator does not return to its initial length, which has extended over the course of the cycling by something like 5%, so the actual range of motion is not only smaller, but because the resting length has

increased, the 47% number would also be lower if taken as a percentage of the actual length at that point in time, as opposed to some previous resting length.

Pg. 8 – You need to spend more time discussing the decrease in achievable elongation. It would have been very sensible to have performed more cycles on the actuators to see if the behaviour saturates, but at the very least you need to discuss possible damage mechanisms that may be affecting performance.

Pg. 8 - The actuator was fixed in a cylindrical container to eliminate radial volume expansion during the measurement of blocked force – why was this done? What is the scientific value of reporting a force number from an unrealistic, over-constrained test? The muscle force characteristics should be measured in a more convincing way.

Pg. 12 – an even faster and more severe reduction in achievable displacement is seen in the pipe crawling version, but once again it is not discussed.

Pg. 14 – “A digital camera (Canon EOS 90D) was used to record changes in artificial muscle elongation and perform statistical calculations using mathematical methods.” – “mathematical methods” is not a useful description of a method, as it tells us nothing about what was actually done!

Answers to Comments by Reviewers

Reviewer #1 (Remarks to the Author):

This study presents a miniaturized, untethered McKibben muscle for soft robots, driven by photothermal-induced liquid-gas transformation. The incorporation of carbon particles into the silicone rubber material facilitates a self-shielding effect, enabling spontaneous core-shell formation under UV light. This streamlines manufacturing and reduces actuator size. By adding a restricting layer of PTFE fiber membrane, the radial expansion of the artificial muscle is effectively suppressed, thereby achieving anisotropic driving ability. After the material and performance characterization, three untethered McKibben muscle functions are demonstrated: a climber on flat plate, a worm-inspired soft robot wriggling in tube, and a gripper consisted of three independent artificial muscles. The untethered artificial muscle presented in this work has potential in enhancing high-force soft robotic applications, with its unique fabrication method enabling downsizing of liquid-gas phase transition-driven actuators.

However, the novelty, completeness and importance of the current submission falls short of the standards of Nature Communications. Addressing and clarifying the following aspects will be crucial before considering publication elsewhere:

The soft actuator introduced in this paper relies exclusively on a low-boiling-point liquid (25°C) as its liquid-gas phase change working medium. However, the actuator does not demonstrate any advantages in terms of operational capabilities, driving performance, size, or structural complexity when compared to previous publications focused on liquid-gas transmission actuators. Furthermore, it's worth noting that the manual winding of constrained fibers contributes to the intricacy of the manufacturing process, thereby limiting the potential for achieving high-precision production and large-scale assembly. This limitation is evident from the prevalent use of sample sizes at the centimeter scale. More efficient designs for anisotropic elongation and force output have already

been presented (“Wireless soft actuator based on liquid-gas phase transition controlled by millimeter-wave irradiation.” IEEE Robotics and Automation Letters, 2020).

Simultaneously, the manuscript neglects to provide information regarding the energy density of the high-power light source, giving rise to concerns about the actuation efficiency. Another notable concern pertains to the stability of the driver's performance, stemming from the utilization of a working fluid with a boiling point closely aligned to room temperature. This decision introduces considerable uncertainties regarding performance reliability, and no obvious advantage has shown considering the high temperature applied to stimulate the actuator. While the proposed fabrication method showcases intriguing elements, its genuine potential remains unexplored due to the absence of comprehensive analysis, theoretical foundations, and adequate characterization.

>>> **Re:** First of all, we would like to express our gratitude for the time and effort you have put into reviewing our manuscript and suggesting improvements, which is of paramount importance to us. We understand your concerns about the performance of the drives described in the manuscript, and taking your feedback into account, we are committed to improving the quality of the manuscript and answering your questions about the details of the experiment.

1. The manuscript emphasizes "light-driven" yet lacks critical light power density details throughout. To ensure accuracy and readability, please provide comprehensive disclosure of experimental processes and results in the main manuscript and supplementary information

>>> **Re:** We appreciate your comments on the lack of information. We are aware that the absence of optical power data raises concerns about drive performance. The light intensity we used throughout the experiment ranged from 100 mW/cm² to 150 mW/cm², and the optical power data from each experiment have been added to the manuscript.

2. In light of the promising driving performance achieved by the proposed soft actuator, a more comprehensive comparison with relevant reported works in the field is needed. For instance, Tang's work (Wireless miniature magnetic phase-change soft actuators, Adv. Mater., 2022) demonstrates a substantial output force and working capacity, Kurumaya's work (Design of thin McKibben muscle and multifilament structure, Sensors and Actuators A: Physical, 2017) shows a pneumatic

McKibben muscle design with comparable size, as well as Cacucciolo's work (Electrically-Driven Soft Fluidic Actuators Combining Stretchable Pumps With Thin McKibben Muscles, *Frontiers in Robotics and AI*, 2020). A thoughtful discussion that contextualizes the proposed soft actuator's performance against such benchmarks will provide valuable insights into its potential and unique attributes.

>>> **Re:** We appreciate you for supplementing the relevant literature that we may have missed, and hope to explain to you the differences between the literature and the author's research. Compared to Soichiro Ueno's work ("Wireless soft actuator based on liquid-gas phase transition controlled by millimeter-wave irradiation." *IEEE Robotics and Automation Letters*, 2020), our work shows the possibility of further reducing the size of the device and demonstrating its function in the application scenario. Kurumaya and Cacucciolo's work on the thin McKibben has been well completed, but due to the limitations of energy devices such as powerful power supplies and air pumps, it is difficult to further reduce the size of the actuator. Therefore, the above method of external air pump and high voltage power supply is contrary to our concept of wireless drive. Tang et al. 's work shows a wireless magnetically driven miniature elastomer that uses phase transitions in a low-boiling liquid to complete the expansion process. They make miniaturized drives by manually making cavities and injecting liquid, which is undoubtedly cumbersome. In our work, by pouring the pre-polymerization liquid into the mold and direct photocuring molding, the elastomer containing a low boiling point liquid can be prepared more simply and quickly, which greatly reduces the production time. And the function of our miniaturized actuator is not simply to expand, but to convert from radial expansion to axial elongation using the winding of a fibrous membrane, mimicking the elongation and contraction of biological muscles during movement.

In summary, our work is to realize light-driven wireless control on the basis of the pneumatic actuator, which retains the high execution power of the pneumatic actuator and avoids the burden of heavy equipment with the help of light. This enabled our work to finally miniaturize the pneumatic actuator, increasing its maneuverability and performing tasks in different application scenarios.

3. The manuscript intriguingly explores a low-boiling point liquid-based actuator requiring a relatively high driving temperature. I'm confused by the mismatching in temperatures of the liquid

boiling point and the heating process, and encourage a thorough investigation into the interplay between temperature, inner pressure, and elongation. Such an exploration could shed light on the underlying physics and mechanisms governing this behavior, potentially leading to insights for optimizing the actuation process.

>>> *Re:* We appreciate your guidance for further discussion and questions about the details of the experiment. We realize that the boiling point of a low-boiling liquid may not be clearly highlighted in our manuscript. In the experiment, the boiling point of the low-boiling liquid we used is 33°C, but the light irradiation is an external heating method, which will be blocked by the driver itself during the heating process, not only that, in the process of rising the temperature of the driver surface, the internal cavity also obstructs the heat transfer. Therefore, the upper part of the driver directly exposed to light has a higher temperature and there is a temperature difference with the self-occluded part, which can be clearly distinguished from the newly provided infrared image in Fig. S8. Therefore, in order to ensure the drive efficiency in the previous experiment, we used the overheating scheme. Limited by the experimental conditions, we cannot accurately calculate or measure the internal pressure change of the driver. Therefore, we measured the blocking force of the driver with different elongation (liquid content is 23%). Obviously, the blocking force decreases continuously during the process of driver elongation. When the force is 0, the elongation of the driver reaches the maximum. From the measured data, it can be seen that the elongation is positively correlated with the temperature. In the whole driving process, when the driver begins to extend, the blocking force will decrease, and the increase of elongation is negatively correlated with temperature.

4. The manuscript reveals an actuation process that, while promising, exhibits notable inefficiencies in terms of duration and speed (8 minutes for one cycle of driving and 40 minutes for achieving a

moving distance of 50% of the body length). To advance the practical utility of the proposed actuator, I recommend delving into potential structural design improvements that could enhance actuation efficiency. This could involve exploring innovative configurations or materials that facilitate faster and more effective actuation.

>>> *Re:* The drive efficiency of the tube driver is reduced by the friction in the tube, and in this application scenario, we use a single light source to manually illuminate the three parts of the driver to control the drive. So the uncertainty of the time control makes the drive efficiency unsatisfactory. Although the three parts structure we designed can be well adapted to the pipeline, it still has defects in precise control. In order to reduce the friction caused by the pipeline structure, we apply silicone oil on the surface of the driver as a lubricant. Surely, as your kind suggestion, the actuation can be faster and more effective after exploring innovative configurations or materials. That might be the chief aim of our subsequent research. For example, we can make the feet into annular expansion belts (manufacturing by mold) fixed at both ends of the body, which can not only reduce the contact area between the driver and the pipe, reduce friction, but also reduce the displacement loss of the body, so as to improve the movement efficiency.

5. The self-shielding effect of UV light demonstrated in the fabrication process is noteworthy for simplifying manufacturing, and endows the possibility of more complex 3D shaped actuators with one-step fabrication. However, the manual wrapping of the PTFE fiber membrane (Line 63) appears labor-intensive. As a suggestion, the authors might consider optimizing this step or exploring alternative fabrication strategies that further emphasize the advancement in manufacturing techniques.

>>> *Re:* We appreciate your guidance on the future development of the drive. The PTFE fiber film we used is a commercial film that can be purchased. When wrapping the drive, we only need to wrap it once because the product film itself has a single fiber orientation. For relatively complex three-dimensional structures in the future, electrospinning spinneret might be a good choice instead of manual winding to improve production efficiency.

6. Demonstrations highlighting the actuator's capacity for generating substantial output force would be invaluable in showcasing its potential applications. This could include scenarios where the

actuator's force output is particularly advantageous, underscoring its practical utility and setting the stage for broader adoption.

>>> **Re:** We understand reviewers' concerns about output force application scenarios. In order to better show the application prospect of the output force, we designed a light-controlled thruster: We made a driver with $L=3.6\text{cm}$ and $d=9\text{mm}$, fixed its left side and placed a 100g weight on the right side. Under light irradiation, the driver shifted the weight to the right side by 2.1cm. The maximum static friction force between the weight and the surface of glass has been measured to be 0.2N. And we added pictures to the Fig. S7.

7. The manuscript introduces a significant advancement in actuator downsizing. To underscore this accomplishment, I suggest contemplating the feasibility of further miniaturizing the actuator to submillimeter dimension. Such exploration could underscore the uniqueness of the proposed fabrication method compared to existing works and highlight its potential for even smaller-scale applications.

>>> **Re:** Honestly, our actuator can hardly be miniaturized into submillimeter range following the current strategy, since we formed a cavity containing a low-boiling liquid by using carbon nanoparticles to block the UV, so further reducing the size means we need to compress the space in which the shell or cavity exists. However, the reduction of the cavity affects the drive effect, and the smallest effective drive cavity volume we tried was around 6.5mm^3 (estimated cavity diameter 1.4mm, length 4.2mm).

8. How did the shell thickness shown in Fig. S2 measured? Whether the thickness varies between different fabrication batches?

>>> **Re:** Thank you for the inquiry about our measurement techniques and the differences in film

thickness between individual drivers. We measured the thickness of the film with a thickness gauge. The film is indeed non-uniform, about 100 microns, which may be caused by the lighting mode of the handheld UV lamp, and improving the lighting mode such as ring UV curing may be beneficial to the uniformity of the film thickness. However, the uniformity of film thickness has no obvious effect on the overall performance of the driver. And we have explained this part in the revised manuscript.

9. Long-term stability remains a critical aspect for practical deployment especially when the phase change liquid holds a boiling point of 25°C. The internally depressed sample in Fig. 2G indicates the severe leak of contained low-boiling point liquid during the fabrication and driving process. Also, 8 times elongation test is inefficient to prove the high reversibility (Line 147). How long does the actuator is expected to be working properly, could the authors comment on this?

>>> **Re:** Thank you very much for this suggestion. In order to prove the reversibility of the drive, we have added the recovery experiment, using the same sample, 20 reversible elongation was completed in 68 mins, and the average elongation was maintained at 36.4%. The optical power density of xenon lamp in the experiment is 100 mW/cm². The increased recovery experimental data and optical power density were also added to the manuscript.

Indeed, the possibility of liquid leakage does exist due to the low boiling point of the liquid as 33 °C and the overheating because of the slow heat-transfer rate. However, this problem can be easily solved by driving the actuator into the low boiling point liquid to restore its performance.

For example, in a specific application scenario (tube crawling), the drive can work normally for more than one hour. Although the problem of leakage seems to be serious, the performance of the drive can be restored by soaking it in a low-boiling liquid. After it completely loses its driving ability, we soak it in a low-boiling liquid, and after soaking for four hours, it can recover about 80% of the driving performance. In Fig. S9, we added the comparison between the elongation of the sample in the initial state and that after 4h soak in the liquid.

10. In the introduction section (Lines 51-53), I suggest redirecting your focus from pneumatic devices to the domain of remotely controlled miniaturized soft robots/actuators. This adjustment is recommended considering the extensive body of existing literature in this field. It is advisable to avoid delving into an extensive and redundant historical account of pneumatic McKibben muscles, as this could potentially dilute the engagement of both your time and the readers' attention. Furthermore, in Lines 55-62, you aptly outline the developmental trajectory of liquid-gas phase transition-based soft actuators. However, the introduction would benefit from the inclusion of pressing and contemporary challenges currently encountered within this field (点明现存挑战). Presenting these challenges succinctly and directly will enable a more engaging introduction, allowing readers to grasp the distinctive contribution of your work and its targeted approach to addressing a specific problem.

>>> **Re:** We appreciate the reviewer's guidance on the foreword. We have deleted part of the description of McKibben and added the following content: "Nowadays, the use of phase change liquids for wireless gas drive has become the choice of more researchers. In order to obtain greater deformation, researchers often choose rubber with high permeability to low boiling point liquids as a carrier, which will lead to liquid leakage. Therefore, the sustainable use of devices will become an issue."

11. The data plots in Fig. 2C, E reveal a notable instability in the elongation of the actuator driven by phase change, characterized by a considerable variation range. It would greatly enhance the manuscript's depth to delve into the potential factors contributing to this observed phenomenon. Furthermore, an insightful discussion of possible strategies or modifications that could be implemented to mitigate this instability would provide readers with valuable insights into enhancing the actuator's consistency and reliability.

>>> **Re:** Since the PTFE film itself does not have adhesion, the fixation of the film becomes a key factor limiting the radial expansion of the driver. After wrapping the PTFE film, we applied a prepolymer solution to its outer layer and cured it as a holding measure for the film. However, only relying on the outer layer fixation cannot make the driver obtain a more efficient radial binding force, that is, during the drive process, the stress at the interface of the PTFE film will exceed the bearing capacity of the fixed layer, resulting in a rapid release of stress in a small range, resulting in a loss of elongation or damage to the device. In order to optimize the stability of the driver elongation, we tried to apply a prepolymer solution to the PTFE overlapping interface and cure it, using it as an adhesive to bind the PTFE membrane interface. This may cause uneven force on both sides of the drive, making it bend, but it is conducive to stability. Explanations and discussions of experimental errors have been added to the manuscript.

12. In Lines 125-127, the conclusion drawn regarding the relationship between particle increase and elastomer membrane airtightness, as indicated in Fig. 2C, warrants further elucidation. The current presentation of the data may not fully substantiate this connection.

>>> **Re:** We can further explain this problem from two aspects. First, when the outer layer is photocured, the doped carbon powder, as a solid particle, does not participate in the chemical crosslinking in the prepolymerization solution. Therefore, excessive carbon powder will cause the structure of the polymerized elastomer to be loose and the air tightness to be weakened. Secondly, the absorption of ultraviolet light by the carbon powder will affect the thickness of the polymer film of the driver, and the increase of the carbo content will reduce the thickness of the polymer film, which also leads to the reduction of the air tightness of the driver to a certain extent. And the relationship between particle increase and the air tightness of elastomer membranes has been added to the manuscript.

13. Detailed discussion is needed in Discussion Section.

>>> **Re:** According to the opinions, we made a further analysis of the uniqueness of our work in the field, and then put some important issues in the discussion section for detailed discussion. For example, the further reduction of size, the improvement of air tightness and the possibility of exploring more functional applications. And following your above suggestion, we have inserted the relative discussion in the certain paragraphs.

14. What does “when the container expands indefinitely” mean (Line 128)?

>>> **Re:** We are sorry for the wrong expression. The sentence has been changed into “The increase in fluid content significantly increases the volume of the artificial muscle after expansion.”

15. What does “the elongation began to increase gradually, reaching a maximum of ~460s” mean (Line 142)?

>>> **Re:** It means “the actuator began to elongate and reach an equilibrium in 460s”. And we have revised the manuscript.

16. Ensuring consistent abbreviation usage is pivotal for reader comprehension. In Line 64, it is advisable to introduce the full name before employing an abbreviation to enhance clarity upon its initial appearance.

>>> **Re:** The author is grateful to the reviewer for the format modification. The abbreviations in the article have been revised and checked.

17. Please mark the liquid drople in Fig. S1, it's hard to identify useful information from the figure.

>>> **Re:** Thank you very much for this suggestion. We have revised it in the manuscript.

18. Also, what kind of mixing strategy was employed to make the droplet distributed uniform in the prepolymer with a diameter of around 3 μm ? The disclosure of necessary experimental steps is essential.

>>> **Re:** We used mechanical agitation as a means of dispersing low boiling point liquids with a

rotational speed range of 235 rpm. We have added the dispersion method to the manuscript.

19. Please provide the data of applied temperature in Fig. 2C (Line 120) to bolster data transparency.

>>> **Re:** In Fig. 2c, we use light as a heating method. Since the maximum elongation is a limit value, we did not measure the temperature in the overheating scheme, but the temperature is positively correlated with the light intensity. Therefore, we added the description of light intensity in the manuscript to describe the experiment more accurately.

20. To enhance the manuscript's organization, providing explicit references to specific figures within the supplementary information using designations like "Fig. S4" instead of the more general "Supplementary figure" would enhance navigability for readers. (Line 268).

>>> **Re:** The author is grateful to the reviewer for correcting the manuscript formatting errors, We have made corrections according to the journal format requirements.

21. When characterizing the elongation of the actuator, discussing any observed changes in the actuator's radius concurrently would provide a more comprehensive understanding of how these properties correlate.

>>> **Re:** Thanks for pointing out this important aspect, we recognize the need for a fuller explanation and discussion of deformation properties. In Fig. 2b, we compare the axial and radial elongation of a PTFE wound and a non-PTFE wound drive. We can see from the image that the radial elongation of the drive varies by up to 63% before winding, while the axial elongation is only 17%. After wrapping the PTFE film, the radial elongation of the drive is reduced to 13% and the axial elongation is increased to 47%. Therefore, we can conclude that the winding of the fiber reduces the radial elongation by 50% and increases the axial elongation by 30%. In this way, we succeeded in changing the main direction of the deformation from radial to axial. Therefore, our winding method effectively increases the structural constraints in a single direction to transform the original radial expansion into axial elongation, which is similar to the deformation mechanism of fiber reinforcement.

22. Could the author comment on how to improve the fabrication process to meet the needs for

massive production

>>> **Re:** The authors thank you for raising the possibility of future large-scale manufacturing of drives. According to the existing production process, we mainly use hand winding method to process the sample. This does raise concerns about large-scale industrial preparation. What we need to explain is that the initial elastomer that is not wrapped in PTFE film can be manufactured on a large scale by means of molds, as long as these molds are transparent and can ensure the passage of ultraviolet light. As for the winding part of PTFE, the winding method we currently choose is to use a commercial PTFE film for single-circle winding, because the fiber orientation of the membrane itself, so we can directly put it into use, only need to wrap and fix along the fiber orientation, we can realize the radial constraint of the cylindrical drive. In the Fig. S8 we provide optical images of PTFE films. This method is not complicated, even the smallest size we can quickly complete the winding. For further large-scale manufacturing, especially for the mass production of miniaturized drives, we can first prepare long and thin elastomers to wrap around the whole, and then cut different lengths according to the need. Finally, the intercepted miniaturized actuators can be quickly sealed by the photocuring properties of the polymer.

23. How does the UV light density and the operating time influence the thickness of the shell? Will the uncured prepolymer gradually cure during the actuation process or after a long time.

>>> **Re:** We appreciate the reviewers' questions about the details of the experiment. Ultraviolet light intensity and time will indeed affect the thickness of the shell. In the experiment, we used a handheld ultraviolet lamp without adjusting the light intensity, and only considered the influence of light time on the shell thickness after fixing the light distance. This part of the experiment has been added in the revised manuscript. In addition, the chemical properties of the prepolymer itself are relatively stable, and it will not be further cured even if it is irradiated by xenon lamp under the cover of carbon black and PTFE film. It will maintain stability for a long time in the visible light environment under occluded conditions. Elongation tests after 5 months showed that the sample still had driving performance.

24. A comprehensive rewrite of the abstract is recommended to concisely emphasize the unique contributions and novelty of this work. Furthermore, a meticulous language polish throughout the

manuscript will ensure clarity and precision, enhancing the presentation of your findings.

>>> **Re:** The abstract has been revised:

Pneumatic artificial muscles can move continuously under the power support of air pumps, and their superior flexibility also provides the possibility for applications in complex environments. However, in order to achieve operation in confined spaces, the miniaturization of artificial muscles becomes crucial. Since external attachment devices greatly hinder the miniaturization and use of artificial muscles, we propose a light-driven approach to get rid of these limitations. In this study, we report a miniaturized fiber-reinforced artificial muscle based on mold editing, capable of bending and axial elongation using gas-liquid conversion in visible light. The minimum volume of the artificial muscle prepared using this method was 15.7 mm³ (d = 2 mm, l = 5 mm), which was smaller than those of other fiber-reinforced pneumatic actuators. In addition, these light-driven artificial muscles can move continuously on flat surfaces and in pipes. This research can promote the development of non-tethered pneumatic actuators for rescue and exploration, and create the possibility of miniaturization of actuators.

Reviewer #2 (Remarks to the Author):

This manuscript presents a novel approach by proposing a miniaturized, untethered McKibben Muscle powered by an internal phase change material, which is activated by UV light. While the study provides valuable insights, I have several key comments that need addressing.

>>> **Re:** We appreciate for your recognition in the drive method and miniaturization, and your valuable comments have played a key role in further improving the quality of our manuscripts. Regarding the reviewer's concern about the safety of ultraviolet light as a driving light source, we considered that it might be caused by unclear descriptions in the manuscript. In fact, our UV light source is only used as a means of curing and will not participate in the driving process.

1. Relevance of UV Activation: The primary appeal of the work lies in its attempt to address the challenge with bulky pumps in pneumatic actuators – a fact I acknowledge and appreciate. However, I question the practicality of using UV light as a driving mechanism. The abstract suggests

applications in "miniaturized medical equipment, rescue, and exploration", but it remains unclear how UV light would be suitable or practical for all these contexts. Could the authors illustrate with a specific scenario? While the manuscript emphasizes the off-board energy source, it's pertinent to remember that this energy still needs to be supplied for the actuator to function. It means the lamp has to be around. Additionally, the penetration depth of UV light in various practical situations should be discussed.

>>> **Re:** We really appreciate your advice, and we need to explain the light source used for the drive. Our drive is achieved by simulating sunlight from xenon lamps, UV is only used in the fabrication process to solidify our drive. We have supplemented the information and intensity of the light source at different places in the article to provide an accurate description. The use of simulated sunlight lays the foundation for use under natural conditions. In the future application scenario, we mentioned small medical devices, which are not easy to achieve for the current research. We will remove the content to give a more objective picture of the future of artificial muscles. The use of xenon light source is our initial attempt to achieve daylight drive, and in the future we hope that the drive can realize its function in sunlight without space constraints.

2. Light Intensity Clarification: The manuscript seems to have overlooked the specifics regarding the intensity of the UV light used. While it mentions the employment of a Xenon lamp, there's no elaboration on its intensity. It's important to understand if the impressive performance of the actuator is primarily due to high light intensity or the inherent design of the actuator. This distinction is crucial as many locomotion demonstrations seem to be driven by the light's direction and intensity rather than the actuator's intrinsic properties.

>>> **Re:** We appreciate your revision suggestion. We are aware of the confusion caused by the absence of optical data. The intensity of the UV lamp is 150mw/cm², and the intensity of the xenon lamp is 100 mw/cm²-150 mw/cm². And we've added the data to the manuscript.

The movement and deformation of our drives are based on the intrinsic properties of the material and the unique design, the light only provides the energy, and in the future, the demand for the operation of the optical equipment can be improved by optimizing the material properties. Of course, the realization of specific mode of motion also requires the corresponding optical operation mode, and it is also based on the structural design of the material, but in the final analysis, it is the basic

properties of the artificial muscle to give the ability. Complex movement through basic deformation and some structural design is also a way for biological muscles to achieve function.

3. Comparison with Prior Work: While the paper makes reference to existing literature (line 60), I believe a more comprehensive review and comparison are warranted. More crucially, the manuscript should elucidate why its approach and results are superior or distinct from previously published studies, especially in those application scenarios mentioned.

>>> **Re:** We appreciate your attention and suggestions in related fields. In the introduction part of the manuscript, we have added some comparisons of existing light-driven small actuators to the preface of the manuscript. Compared with the work of other light-driven actuators, our work hopes to get rid of the simple bending drive mode and realize more complex motion by using the high execution force of pneumatics and deformation of different dimensions. At present, visible light control is rare in the research of pneumatic actuators, and we have made exploration and attempts in miniaturization of pneumatic actuators. Our mold manufacturing method can create different three-dimensional shapes to adapt to different environmental needs in future applications.

In conclusion, while the manuscript offers an intriguing concept, it would greatly benefit from addressing the aforementioned concerns. There's a plethora of work in the domain of light-controlled miniature robots, and this study should firmly establish its unique contribution and relevance in the field.

Reviewer #3 (Remarks to the Author):

Overall, this is an interesting and reasonably well written paper. The proposed concept for a self-contained, light driven fluidic muscle actuator appears to be effective and simple. The description of the concept and the associated methodology is fairly well presented, and the different demonstrations showing the working of the concept are insightful. Some of the details of the technical work are a bit weaker though, and some issues present in the results are not adequately

discussed. In particular, the significant loss in performance seen in initial cycling with both the muscles themselves and the tube crawler device need further discussion. Ideally a more thorough running in would have been performed to see if the loss in deformation saturates to a consistent level, after which the data for presenting should have been taken. It would have been particularly useful to see the force versus displacement behaviour characterized to better understand the ability of these devices to do work. The bending walker and tube crawler are only carrying their self-weight, and since force/energy capability aren't measured, we can't make any conclusions as to their ability to do meaningful work in the real world. A single force number is presented, but it is completely unclear to this reviewer why they chose to put the muscle into a separate external tube (thereby artificially and unrealistically constraining its diameter) when taking this measurement, and as a result this number doesn't seem to mean anything. Ultimately though, the work is quite interesting and if these minor weaknesses can be corrected then I believe it is worthy of publication. Some detailed comments follow :

>>> **Re:** We appreciate for your recognition and encouragement of the light-driven fluid muscle, which gives us the motivation and determination to further improve the quality of our manuscripts. In response to your feedback, we have added the analysis and discussion of application scenarios in the manuscript, and added the application scenarios of output force.

Pg. 3 – “classic structures of McKibben muscles develop” - “develop” is misspelled and not needed

>>> **Re:** The author is grateful to the reviewer for correcting the language errors in the manuscript, and we have made corrections in the manuscript.

Pg. 4 – The moulding technique and partial curing to create an outer membrane and an internal working fluid is clever, but I am wondering about the long term durability – in particular the likelihood of continued curing over time and a slowly thickening wall.

>>> **Re:** We take seriously the reviewer's confusion about the durability of the drive. To illustrate the durability of the sample, we show below a bendable sample (gripper part) preserved in a low boiling liquid for 7 months. Since the outer seal was slightly removed during this period, we reapplied the seal and observed its deformability under light. As shown in the figure, the sample still

retains good bending characteristics.

Although our sample will not continue to cure during the process of optical drive, the outer layer of the sample will inevitably fall off due to long-term storage, so after the sample is put back into use, we need to strengthen the seal with the same method to avoid unnecessary loss. During storage, because the sample is always in contact with the air, the volatile components present inside the sample will be lost, resulting in the remaining part of the fluid adhering to the inner wall, but this will not affect the subsequent use.

Pg. 7 – The quoted maximum elongation of 47.07% after 8 cycles is not a fully accurate description, as the actuator does not return to its initial length, which has extended over the course of the cycling by something like 5%, so the actual range of motion is not only smaller, but because the resting length has increased, the 47% number would also be lower if taken as a percentage of the actual length at that point in time, as opposed to some previous resting length.

>>> **Re:** We appreciate the reviewers for the insightful suggestions on statistical methods. According to the comments of the reviewers, we adjured the calculation method of elongation in the repeatable experiment and increased the number of cycles. And it has been added to the manuscript.

Pg. 8 – You need to spend more time discussing the decrease in achievable elongation. It would have been very sensible to have performed more cycles on the actuators to see if the behaviour saturates, but at the very least you need to discuss possible damage mechanisms that may be affecting performance.

>>> **Re:** In order to explore the repeatability of the sample elongation, we increased the number of cycles to 20, carried out the elongation and recovery experiments continuously, and recorded the elongation of each cycle according to the statistical method suggested by the reviewers. According to statistics, the average elongation of 20 cycles is 36.4%. With the increase of the number of cycles, there is a loss of elongation, which is something we need to improve. The mechanism of damage is mainly considered gas leakage. First, the pores on the surface of the sample expand after elongation, and the high temperature at this time speeds up the rate of gas seepage. Secondly, due to the addition of nano carbon powder when curing the elastomer, the whole elastomer is not tightly cross-linked, and the structure becomes relatively loose during several stretches and cannot maintain the initial state. In order to reduce the effect of gas leakage on the elongation, the sample with damaged elongation was immersed in a low boiling point liquid for replenishment. By elongation verification in Fig. S9, we prove that This method is sufficient to restore the sample to its original driving effect as long as it is soaked for enough time. The discussion of the mechanism of damage has been added to the manuscript.

Pg. 8 - The actuator was fixed in a cylindrical container to eliminate radial volume expansion during the measurement of blocked force – why was this done? What is the scientific value of reporting a force number from an unrealistic, over-constrained test? The muscle force characteristics should be measured in a more convincing way.

>>> **Re:** We are aware of the importance of the issues raised by the reviewers and would like to explain them. When measuring the output force, we incorrectly described the measurement scene. It is accurate to say that in the process of measuring the axial output force, we must apply a radial restraint (a cylindrical sleeve that can wrap the muscle) to the artificial muscle, otherwise the muscle will bend during the process of elongation, which will affect the measurement of the output force. In order to better show the application prospect of the output force, we designed a light-controlled thruster: We made a driver with $L=3.6\text{cm}$ and $d=9\text{mm}$, fixed its left side and placed a 100g weight on the right side. Under light irradiation, the driver shifted the weight to the right side by 2.1cm. The maximum static friction force between the weight and the surface of glass has been measured to be 0.2N. And we added pictures to the Fig. S7.

Pg. 12 – an even faster and more severe reduction in achievable displacement is seen in the pipe

crawling version, but once again it is not discussed.

>>> **Re:** We really appreciate your suggestion, which is very useful to us. And we have added this part of the discussion to the manuscript. The further discussion of displacement reduction in the pipeline crawling version is as follows: First, in the pipeline crawling experiment we used a light mode of delayed segmenting irradiation, and the displacement generated during the anchoring may be lost due to the delay of the optical operation. After the anchoring of the front foot is completed, the illumination of the front foot is prematurely cancelled during the waiting time for the body to retract, so that the anchoring is cancelled when the body is not fully recovered. Therefore, the body loses its anchor point and begins to retract toward the center, driving the front foot to retreat, resulting in the loss of displacement. Secondly, the friction in the pipeline will also have a certain impact on the displacement.

Pg. 14 – “A digital camera (Canon EOS 90D) was used to record changes in artificial muscle elongation and perform statistical calculations using mathematical methods.” – “mathematical methods” is not a useful description of a method, as it tells us nothing about what was actually done!

>>> **Re:** Thanks for your correction, we have modified the relevant statement. For the motion video, we use the ImageJ program to carry out statistics and analysis of the coordinates of the drive point.

REVIEWERS' COMMENTS

Reviewer #1 (Remarks to the Author):

The authors have addressed all my concerns.

Another correction should be considered before publishing: the scale bar is necessary for all supplementary videos.

Reviewer #3 (Remarks to the Author):

The edits made to the paper during the revision process are adequate from my perspective.

Answers to Comments by Reviewers

Reviewer #1 (Remarks to the Author):

The authors have addressed all my concerns.

Another correction should be considered before publishing: the scale bar is necessary for all supplementary videos.

>>> *Re:* We are very grateful for your recognition of our last reply. According to your suggestion, the scale bar has been added to the video. Finally, we would like to thank you for taking time to review our manuscript, which greatly improves its quality.

Reviewer #3 (Remarks to the Author):

The edits made to the paper during the revision process are adequate from my perspective

>>> *Re:* Thank you very much for your approval of the revised manuscript, your suggestions have helped us to improve the manuscript quality.